# Continual Learning of Control Primitives: Skill Discovery via Reset-Games

**Kelvin Xu**[*1], **Siddharth Verma**[*1], **Chelsea Finn**[2], **Sergey Levine**[1]
[1] UC Berkeley, [2] Stanford University

## Abstract

Reinforcement learning has the potential to automate the acquisition of behavior in complex settings, but in order for it to be successfully deployed, a number of practical challenges must be addressed. First, in real world settings, when an agent attempts a tasks and fails, the environment must somehow "reset" so that the agent can attempt the task again. While easy in simulation, this could require considerable human effort in the real world, especially if the number of trials is very large. Second, real world learning is often limited by challenges in exploration, as complex, temporally extended behavior is often times difficult to acquire with random exploration. In this work, we show how a single method can allow an agent to acquire skills with minimal supervision while removing the need for resets. We do this by exploiting the insight that the need to "reset" an agent to a broad set of initial states for a learning task provides a natural setting to learn a diverse set of "reset-skills." We propose a general-sum game formulation that naturally balances the objective of resetting and learning skills, and demonstrate that this approach improves performance on reset-free tasks, and additionally show that the skills we obtain can be used to significantly accelerate downstream learning. [2]

## 1 Introduction

For reinforcement learning (RL) methods to be successfully deployed in the real world, they must solve a number of distinct challenges. First, agents that learn in real world settings, such as robotics, must contend with non-episodic learning processes: when the agent attempts the task and fails, it must start from that resulting failure state for the next attempt. Otherwise, it would require a manually-provided "reset," which is easy in simulated environments, but takes considerable human effort in the real world, thereby reducing autonomy – the very thing that makes RL appealing. Second, complex and temporally extended behaviors can be exceedingly hard to learn with naïve exploration [25, 24]. Agents that are equipped with a repertoire of skills or primitives (e.g., options [43]) could master such temporally extended tasks much more efficiently. These two problems may on the surface seem unrelated. However, the non-episodic learning problem may be addressed by learning additional skills for resetting the system. Could these same skills also provide a natural way to accelerate learning of downstream tasks? This is the question we study in this paper. Our goal is to understand how diverse "reset skills" can simultaneously enable non-episodic reset-free learning and, in the process, acquire useful primitives for accelerating downstream reinforcement learning.

Prior works have explored automated skill learning using intrinsic, unsupervised reward objectives [20, 13, 40], acquiring behaviors that can then be used to solve downstream tasks with user-specified objectives efficiently, often by employing hierarchical methods [11, 43, 12]. These skill discovery methods assume the ability to reset to initial states during this "pre-training" phase, and even then face a challenging optimization problem: without any additional manual guidance, the space of potential

---

[*]equal contribution

[2]code is available at https://github.com/siddharthverma314/adversarial.git

behaviors is so large that practically useful skill repertoires can only be discovered in relatively simple and low dimensional domains. Therefore, we need a compromise – a more "focused" method that is likely to produce skills that are relevant for downstream problems, without requiring these skills to be defined manually.

In this paper, we study how these issues can be addressed together with removing manual resets. While the problems of skill discovery and reset-free learning may at first appear unrelated, we observe that reset-free learning of a given task can be enhanced by access to varied and diverse "reset skills," which force the task policy to succeed from a variety of starting states. At the same time, the skills can be forced to acquire more complex behaviors, through an objective that encourages them to discover states that are challenging for the task policy. This means that the same mechanism that can address the "reset" problem can also serve as a skill discovery method. Performing a single given task can be viewed as a "funneling" process: transitioning from a wide distribution over initial conditions to a narrower distribution over states where a given task has been completed successfully. Reversing this process therefore can provide coverage over a broader range of states, while still keeping this exploration process grounded to situations from which the initial task is solvable.

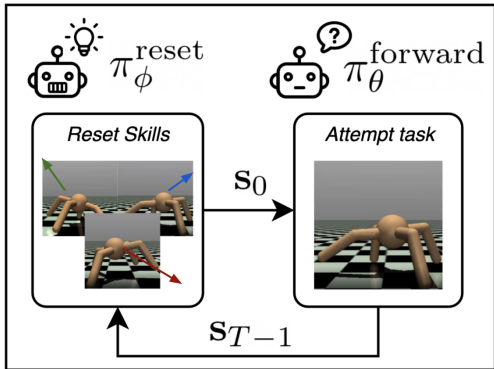

Figure 1: An outline of our approach. A reset policy $\pi_\phi^{\text{reset}}$ provides a state $\mathbf{s}_0$ to a forward policy $\pi_\theta$ to learn a task using a learned skill. Upon completion, the final states, $\mathbf{s}_{T-1}$ is reset with another selected skill. In real world settings, where all learning is continuous and sample efficiency is critical, we present a method that leverages the insight that the non-episodic learning problem can be addressed by learning a set of skills of resetting the system. We additionally show that these skills can be used to accelerate downstream learning.

The main contribution of this paper is a "reset game" that implements this idea as a general-sum game between two players: a task policy player that attempts to perform a given task, and a reset policy player, corresponding to a repertoire of distinct skills, that attempts to perturb the state to make the task harder for the task policy while *simultaneously* producing a diverse set of skills (see Figure 1). The result of this process is a setting where the challenge of learning without resets is instead viewed as an opportunity to learn a diverse set of behavioral primitives, which can then be used to accelerate downstream reinforcement learning, both removing the need for human-provided resets and acquiring skills that can be used for downstream tasks. We show that by grounding the skill learning problem in this way, we can design a method that (1) enables an agent to learn a task without requiring oracle resets, which is a non-trivial requirement in real-world applications, and (2) learns a broader set of skills compared to prior unsupervised approaches, which substantially improves effectiveness on down-stream long-horizon tasks. We demonstrate the efficacy of our approach both on previously studied robotics-themed reset-free RL problems, and tasks that reflect domains studied in prior unsupervised skill learning work, demonstrating both improved reset-free learning performance and improved performance on downstream tasks with hierarchical controllers.

## 2 Related Work

Learning without access to resets has been studied in prior work with a focus on automation, safety, and learning compound controllers [23, 13, 8]. Eysenbach et al. [13] propose to learn a reset controller for safe learning, but assume access to ground truth rewards and an oracle function that can detect resets. Closely related to our work, Zhu et al. [54] learn a perturbation controller to handle the reset-free setting, but importantly does not learn a set of skills. In contrast, we propose to learn a broad set of skills for performing resets, which are "anchored" to a specific forward RL task by requiring that the reset skills produces states that are diverse and challenging. We show that this anchoring ultimately leads to skills more suited for downstream learning, while improving reset-free performance.

Autonomous acquisition of skills is related to the learning with intrinsic motivation, which considers the unsupervised setting where an agent must learn without extrinsic reward [39, 9, 26, 4, 3]. In the direction of curiosity-driven exploration, recent work has used state novelty to reward an agent [5, 44,

33]. Similarly, other work has instead used the empowerment maximization principle [38] to define mutual information based objectives from which to learn behavioral primitives [30, 20, 16, 14, 40]. Our method incorporates unsupervised skill discovery into reset-free learning, building on mutual information methods for skill discovery. In contrast to these prior methods, our approach uses a forward task to implicitly "anchor" the skills. We show experimentally that this allows us to learn varied skills in the reset-free setting, and also produces better skills for downstream task learning.

The goal of acquiring primitives for the purpose of enabling efficient learning on downstream tasks is also a central goal of hierarchical reinforcement learning (HRL), which has been a long-standing area of research [11, 51, 32, 43, 12]. Recent work has investigated deep hierarchical agents [1, 48, 16, 31], building on methods that explicitly design a hierarchy over actions [32, 12], or utilize options [43, 35]. Our work is also motivated by acquiring temporal abstraction through experience, although we seek to do so under a problem setting that is reset-free. Our work can be seen as complementary to prior approaches in HRL, as the skills we acquire can be used by an HRL agent to accelerate downstream learning. We demonstrate this empirically in Section 5.

Central to our method is formulating the problem of acquiring diverse skills as an adversarial two player game. Adversarial games have been proposed in RL with the goal of improving exploration [42, 27], learning goal conditioned policies [36] and generating an automated curriculum [45, 41, 2]. Our work resembles the reset variant of asymmetric self-play [42], though our approach differs in two ways. First, while the reset variant of asymmetric self-play defines "Bob's" learning objective using its ability to reset "Alice's" trajectories, the problem setting is not in fact reset-free as Alice is reset at every episode. Second, our method does not involve setting new goals or use time to completion as a proxy reward, but rather makes use of a task reward to measure which reset states are challenging. Unlike prior methods that formulate an adversarial game around goal reaching [42, 36, 17], our method does not involve setting or reaching goals, but rather uses a single task to focus the forward policy, providing opportunities for the reset skills to learn varied adversarial perturbations. Unlike methods that adversarially vary the environment [6, 50], our method does not require any additional privileged capability to change the environment parameters. Instead, the reset skills use the same action space as the forward policy to discover states from which the original task is difficult. Finally, unlike all of the previously listed methods, our approach enables learning in a reset-free setting – a setting where, as we show experimentally in Section 5, prior methods struggle to learn effectively.

## 3 Preliminaries

Our eventual goal is to devise a reset-free, non-episodic learning framework. We first describe standard episodic RL. An RL problem is defined on a Markov decision process (MDP), represented by the tuple: $\mathcal{M} = (\mathcal{S}, \mathcal{A}, \mathcal{P}_\mathbf{s}, r, \gamma, \mathcal{P}_0)$, where $\mathcal{S}$ is a set of continuous states and $\mathcal{A}$ is a set of continuous actions, $\mathcal{P}_\mathbf{s} : \mathcal{S} \times \mathcal{A} \times \mathcal{S} \to \mathbb{R}$ is the transition probability density, $r : \mathcal{S} \times \mathcal{A} \to \mathbb{R}$ is the reward function, $\gamma$ is the discount factor and $\mathcal{P}_0$ is the initial state distribution. The $\gamma$-discounted return $R(\tau)$ of a trajectory $\tau = (\mathbf{s}_0, \mathbf{a}_0, \dots \mathbf{s}_{T-1}, \mathbf{a}_{T-1})$ is $\sum_{t=0}^{T-1} \gamma^t r(\mathbf{s}_t, \mathbf{a}_t)$. In episodic, finite horizon tasks of length $T$, the goal is to learn a policy $\pi_\theta : \mathcal{A} \times \mathcal{S} \to \mathbb{R}$ that maximizes the objective $J(\pi_\theta) = \mathbb{E}_{\pi_\theta, \mathcal{P}_0, \mathcal{P}_\mathbf{s}}[R(\tau)]$. Here, $\theta$ denotes the parameters of the policy $\pi_\theta$, which are learned by iteratively sampling episodes, where at the end of each episode, a new initial state is $\mathbf{s}_0$ is sampled from the initial state distribution $\mathcal{P}_0$. While such resets can be easily obtained in simulated tasks (e.g., Atari [29]), this is significantly more challenging in real-world physical tasks, as it requires human intervention in the form of manual resets or the availability of a hard coded reset [28, 47, 37, 52, 21].

Our method will simultaneously remove the need for manual resets and, at the same time, learn a varied set of skills by resetting the environment in different ways. To learn this set of skills, we build on previously proposed episodic skill learning methods, which we review here. Following prior work [14, 40], skills can be represented by conditioning the policy on a latent skill index $\mathbf{z}$, which is sampled from $\mathcal{P}(\mathbf{z})$ and kept fixed over a temporally extended period. Every distinct value of $\mathbf{z}$ should correspond to a distinct behavior. One objective for achieving this proposed in prior work [20, 14, 40, 34] is the mutual information (MI) between skills and the states they visit: $\mathcal{I}(\mathbf{s}; \mathbf{z}) = \mathcal{H}(\mathbf{s}) - \mathcal{H}(\mathbf{s}|\mathbf{z})$. Maximizing $\mathcal{I}(\mathbf{s}; \mathbf{z})$ entails maximizing the state entropy (high state coverage) while minimizing conditional state entropy (high predictability for each $\mathbf{z}$). This is typically combined with minimizing the MI between actions ($\mathbf{a}$) and the state/context ($\mathbf{s}, \mathbf{z}$), so that skills are distinguished by the states they visit rather than the actions they take. The resulting

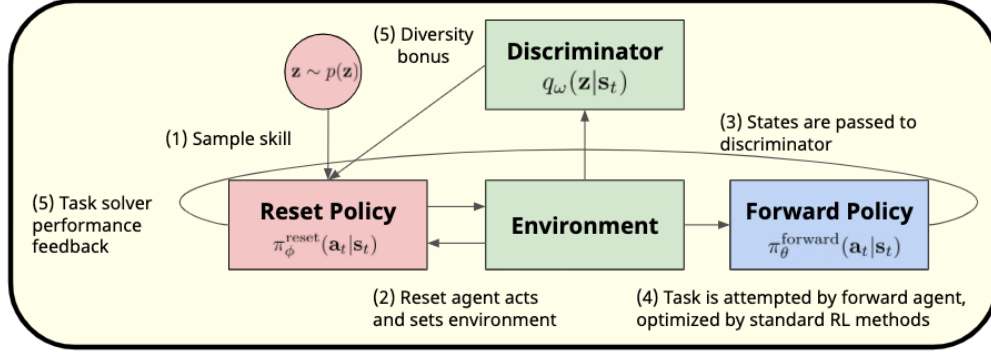

Figure 2: An outline of our approach. Our method first begins by (1) sampling a skill from a prior distribution that is used to condition the reset policy. Next, the reset agent acts in the environment in order to bring the agent to the initial state for the forward policy (2). The states of the reset policy are passed to additionally to a learned discriminator which tries to determine which skill was used to generate the final state (3). Next, the forward policy tries to solve the task (4) and all agents attempt to learn by maximizing their respective objective (5).

objective [14] is:

$$
\begin{aligned}
\mathcal{I}(\mathbf{s};\mathbf{z}) - \mathcal{I}(\mathbf{a};\mathbf{s},\mathbf{z}) &= \mathbb{E}_\pi \left[ \log \frac{p(\mathbf{z}|\mathbf{s})}{p(\mathbf{z})} - \log \frac{\pi(\mathbf{a}|\mathbf{s},\mathbf{z})}{\pi(\mathbf{a})} \right] \\
&\geq \mathbb{E}_\pi \left[ \log q_\omega(\mathbf{z}|\mathbf{s}) - \log p(\mathbf{z}) - \log \pi(\mathbf{a}|\mathbf{s},\mathbf{z}) \right] = \mathcal{G}(\theta,\omega),
\end{aligned}
$$

where we replace $p(\mathbf{z}|\mathbf{s})$ with an approximate learned discriminator $q_\omega(\mathbf{z}|\mathbf{s})$ with parameters $\omega$ to obtain a variational lower bound. We can maximize $\mathcal{G}(\theta,\omega)$ with RL using the pseudo-reward:

$$
r_{\text{skill}}(\pi, q_\omega) = \log q_\omega(\mathbf{z}|\mathbf{s}) - \log p(\mathbf{z}) - \log \pi(\mathbf{a}|\mathbf{s},\mathbf{z}). \tag{1}
$$

The skill learning algorithm alternates between learning the skills with this pseudo-reward and optimizing for a discriminator that is able to discriminate between the skills.

## 4   Learning Skills via the Reset Game

To learn without resets, our approach aims to learn both a *forward* policy $\pi_\theta(\mathbf{a}|\mathbf{s})$ and a set of *reset* skills, which we denote $\pi_\phi^{\text{reset}}(\mathbf{a}|\mathbf{s},\mathbf{z})$. We first describe the optimization objectives for each of the policies. Then, we describe how we can balance these two objectives using a game theoretic formulation. Finally, we instantiate this method and present a practical algorithm, where a single goal-oriented forward policy is reset by multiple different reset skills. While learning multiple skills and learning without resets may at first seem like largely unrelated problems, we make the observation that a sufficiently diverse set of skills can serve to alleviate the need for a manual reset, because different policies (i.e., $\pi_\theta(\mathbf{a}|\mathbf{s})$ and $\pi_\phi^{\text{reset}}(\mathbf{a}|\mathbf{s},\mathbf{z})$) can reset *each other* into different states. By forcing $\pi_\theta(\mathbf{a}|\mathbf{s})$ to succeed from the final states of a range of different reset skills, $\pi_\theta(\mathbf{a}|\mathbf{s})$ becomes proficient at performing the task from many states. By forcing the reset skills to all perturb $\pi_\theta(\mathbf{a}|\mathbf{s})$ in different ways, the reset skills themselves are forced to differentiate and learn various behaviors, making them suitable for downstream learning of more complex tasks. In this way, the problems of learning without resets and learning diverse skills are a natural fit to be addressed jointly.

Our goal is to devise a method such that our policies have the following properties: (1) executing the reset policy allows the forward policy to learn to solve the task with standard episodic RL (i.e., without an oracle reset to the initial state distribution), (2) through repeated resetting, the *reset* policy is able to discover skills that may be useful for downstream tasks.

### 4.1   The Reset Game

We first describe the reset game that represents the core of our method, and then in Section 4.2 extend this to incorporate multiple skills by conditioning $\pi_\phi^{\text{reset}}$ on $\mathbf{z}$. Our proposed reset game enables reset-free learning via a competitive interaction between $\pi_\theta(\mathbf{a}|\mathbf{s})$ and $\pi_\phi^{\text{reset}}(\mathbf{a}|\mathbf{s})$, specified as follows:

$$
\max_{\pi_\theta} \quad \mathcal{J}^{\text{forward}}(\pi_\theta, \pi_\phi^{\text{reset}}), \quad \max_{\pi_\phi^{\text{reset}}} \quad \mathcal{J}^{\text{reset}}(\pi_\theta, \pi_\phi^{\text{reset}}), \tag{2}
$$

where $\mathcal{J}^{\text{forward}}$ and $\mathcal{J}^{\text{reset}}$ are the expected returns, which we define below. Both $J^{\text{forward}}$ and $J^{\text{reset}}$ depend on both $\pi_\theta$ and $\pi_\phi^{\text{reset}}$, since the starting state for each policy is obtained by running the other policy (i.e., $J^{\text{forward}}(\pi_\theta, \pi_\phi^{\text{reset}}) = \mathbb{E}_{\pi_\theta, \pi_\phi^{\text{reset}}, \mathcal{P}_\mathbf{s}}[R(\tau)]$, and vice-versa).

$\mathcal{J}^{\text{forward}}$ is the task objective, which forces the task policy $\pi_\theta(\mathbf{a}|\mathbf{s})$ to learn to perform the given task. $\mathcal{J}^{\text{reset}}$ is an objective that causes the reset policy $\pi_\phi^{\text{reset}}(\mathbf{a}|\mathbf{s})$ to discover states from which it is difficult for the task policy to succeed at the task, while at the same time promoting diverse and varied resets. We provide specific definitions for these objectives in Section 4.2, though a number of different options are possible. Crucially, this is not a zero-sum game. However, due to the fact that the reset policy and task policy must alternate, the reset game has a pre-specified turn order. This form of sequential game is known as a Stackelberg game [49], and is closely connected to the field of bi-level optimization [10]. Unlike general games, these games are known to be solvable (using game theoretic measures) by gradient based learning algorithms [15], even when the game is not zero-sum. Formally, a Stackelberg game is defined as an optimization problem where the "leader" must optimize the following optimization problem:

$$\max_{x_1 \in X_1} \{ f_1(x_1, x_2) \mid x_2 \in \arg\max_{y \in x_2} f_2(x_1, y) \}.$$

The follower optimizes the function $\arg\max_{x_2 \in X_2} f_2(x_1, x_2)$. Taking the reset policy as the leader in our setting results in the following optimization problem:

$$\max_{\pi_\phi^{\text{reset}}} \{ \mathcal{J}^{\text{reset}}(\pi_\theta^*, \pi_\phi^{\text{reset}}) \mid \pi_\theta^* \in \arg\max_{\pi_\theta} \mathcal{J}^{\text{forward}}(\pi_\theta, \pi_\phi^{\text{reset}}) \}. \tag{3}$$

Based on this connection, we can devise a gradient-based algorithm to solve the reset game in Equation 2, so long as the reset policy and task policy learn at different rates (e.g., with the task policy learning faster). We present such an algorithm in Section 4.3, but first we describe how the reset policy can discover diverse skills during the reset.

## 4.2 Mining Diverse Skills from Diverse Resets

Recall that our goal is to allow the forward policy to learn to optimize its task reward $r$, as if a reset mechanism were present, and for our reset mechanism to provide challenging and diverse resets. Towards our first goal, we propose to optimize the forward RL policy from the initial state distribution implied by executing $\pi_\phi^{\text{reset}}$ for $T_{reset}$ time steps. Towards our second goal, we propose to use the skill learning reward $r_{\text{skill}}$ defined in Equation 1 to simultaneously acquire a repertoire of reset skills that are forced to diversify. The underlying mutual information objective encourages high coverage of initial states, but to ensure that these states are challenging for $\pi_\theta$, we add the negative return of $\pi_\theta$ to the objective for each reset skill. Putting these terms in the game defined in Equation 3 results in the following optimization problem:

$$\max_\phi \underbrace{\mathbb{E}_{\mathbf{s}_{t'}, \mathbf{a}_{t'} \sim \pi_\phi^{\text{reset}}} \left[ \sum_{t=0}^{T_{reset}-1} \gamma^t r_{skill}(\mathbf{a}_{t'}, \mathbf{s}_{t'}) - \lambda \, \mathbb{E}_{\pi_{\theta*}} \left[ \sum_{t=0}^{T-1} \gamma^t r(\mathbf{a}_t, \mathbf{s}_t) \right] \right]}_{\mathcal{J}^{\text{reset}}(\pi_\theta, \pi_\phi^{\text{reset}})} \tag{4}$$

$$\text{such that } \theta^* = \arg\max_\theta \underbrace{\mathbb{E}_{\pi_\theta} \left[ \sum_{t=0}^{T-1} \gamma^t r(\mathbf{a}_t, \mathbf{s}_t) \right]}_{\mathcal{J}^{\text{forward}}(\pi_\theta, \pi_\phi^{\text{reset}})}. \tag{5}$$

The hyperparameter $\lambda$ controls the relative importance of the two terms of $\mathcal{J}^{\text{reset}}(\pi_\theta, \pi_\phi^{\text{reset}})$. Intuitively, setting $\lambda = 0$ reduces the objective of the reset controller to prior skill learning methods [14], while $\lambda > 0$ requires the reset controller to reach more challenging terminal states. As the task policy and reset skills improve, they provide a curriculum for one another, with the task policy pushing the skills to reach more distant states, and the skills pushing the policy to be effective from a larger range of initializations. As we demonstrate in our experiments, this results in more complex and varied skills, as well as faster reset-free learning for $\pi_\theta$. To reflect the synthesis of resets and skill learning, we refer to our specific instantiation of the reset game as **Learning Skillful Resets** (LSR).

Running this reset game produces an effective forward policy $\pi_\theta(\mathbf{a}|\mathbf{s})$, as well as a set of skills defined by $\pi_\phi^{\text{reset}}(\mathbf{a}|\mathbf{s}, \mathbf{z})$. The latter can be used to solve more complex downstream tasks in a hierarchical RL

---

**Algorithm 1** Learning Skillful Resets (LSR)

---

1: **Input:** Environment
2: **Initialize:** policy $\pi_\theta$, reset policy $\pi_\phi^{\text{reset}}$, discriminator $q_\omega(\mathbf{s}_t|\mathbf{a}_t)$, prior $p(z)$
3: **for** $N$ iterations **do**
4:     Sample skill $\mathbf{z} \sim p(\mathbf{z})$
5:     # *Set* environment
6:     **for** $t \leftarrow 0 \dots T_{\text{reset}-1}$ **do**
7:         Sample $\mathbf{a}_t \sim \pi_\phi^{\text{reset}}(\mathbf{a}_t|\mathbf{s}_t, \mathbf{z})$
8:         Step env $\mathbf{s}_{t+1} \sim \mathcal{P}_\mathbf{s}(\mathbf{s}_{t+1}|\mathbf{s}_t, \mathbf{a}_t)$
9:         Compute reward for reset controller $r_t^{reset} = q_\omega(\mathbf{s}_t|\mathbf{a}_t)$
10:    **end for**
11:    # *Solve* task
12:    **for** $t \leftarrow 0 \dots T - 1$ **do**
13:        Sample $\mathbf{a}_t \sim \pi_\theta(\mathbf{a}_t|\mathbf{s}_t)$
14:        Step env $\mathbf{s}_{t+1} \sim \mathcal{P}_\mathbf{s}(\mathbf{s}_{t+1}|\mathbf{s}_t, \mathbf{a}_t)$, obtain environment reward $r_t$
15:    **end for**
16:    Update reset policy's final reward $r_T^{reset} = -\sum_{t=0}^{T} \gamma^t r_t(\mathbf{a}_t, \mathbf{s}_t)$
17:    Update $\pi_\phi^{\text{reset}}$, $\pi_\theta$ to maximize respective return using SAC
18:    Update discriminator $q_\omega$ using Adam.
19: **end for**

---

framework, using a higher-level policy $\pi^{\text{hi}}(\mathbf{z}|\mathbf{s})$ to command the skill $\mathbf{z}$ that should be executed at each (higher-level) time step. We demonstrate this capability in our experiments.

### 4.3 Algorithm Summary: Optimizing the Reset Game

We could optimize the task policy and reset skills as per Equation 4 via a bi-level optimization, where the parameters of $\pi_\phi^{\text{reset}}$ are held fixed while $\pi_\theta$ is optimized to convergence. However, in an RL setting, this would require an excessive number of samples. A substantially more efficient method alternates gradient steps on both objectives, with different learning rates. Since the reset game corresponds to a Stackelberg game, gradients-based learning algorithms with respect to these objectives exist that have finite time high probability bounds for local convergence in the case of non zero-sum objectives [15]. Optimizing such a problem can be done approximately by using a two-timescale algorithm, where the leader is optimized at a smaller learning rate [15]. This intuitively results in a reset controller that changes "slower" relative to the forward policy. In our implementation, we approximately optimize $\pi_\theta(\mathbf{a}|\mathbf{s})$ and $\pi_\phi^{\text{reset}}(\mathbf{a}|\mathbf{s}, \mathbf{z})$ using soft actor-critic (SAC) [22].

We outline the pseudo-code of our approach in Algorithm 1. At each iteration, we sample a reset skill $\mathbf{z} \sim p(\mathbf{z})$, execute that skill by following $\pi_\phi^{\text{reset}}(\mathbf{a}_t|\mathbf{s}_t, \mathbf{z})$ to reset to a challenging initial state, then execute $\pi_\theta(\mathbf{a}|\mathbf{s})$ to attempt the task. The experience collected during these trials is then used to update the corresponding policies.

## 5 Experiments

In this section we aim to experimentally answer the following questions: (1) How does our approach compare to prior methods in the reset free domain? (2) How do the skills learned by our approach compare to prior work? Specifically, can we learn better hierarchical controllers using the primitives learned by our approach (LSR)? We first describe our experimental setup, evaluation metrics, and the prior methods to which we will compare. We leave detailed discussion of hyperparameters and environment parameters to the Appendix 6.

**Experimental setup.** To study reset-free learning, we use the three-fingered hand repositioning task proposed by Zhu et al. [54], where the hand must position an object in the center of a bin, starting from any position. This is the most challenging domain considered by Zhu et al. [54] for reset-free learning. We refer to this environment as the `DClaw-ManipulateFreeObject` environment. We follow the evaluation protocol of Zhu et al. [54], using an evaluation metric that measures the final

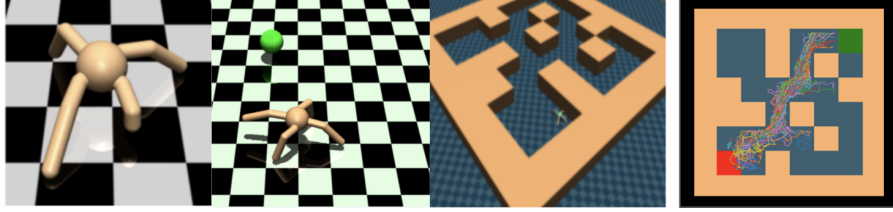

Figure 3: Diagram of the hierarchical locomotion tasks considered in this work. The agent must first acquire locomotion skills in a reset game (left), where the task policy must learn to walk to the origin of the workspace. The skills are subsequently used as the action space for a hierarchical policy, which must learn complex hierarchical navigation tasks (second and third image). The far-right image shows the paths taken by a hierarchical policy using our learned reset skills.

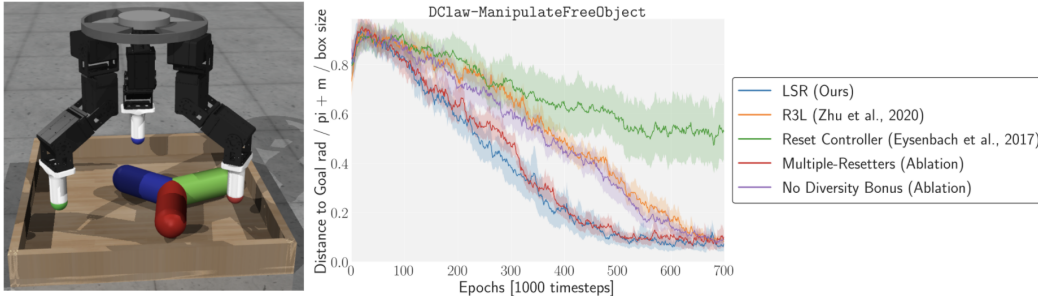

Figure 4: Reset-free learning comparison (lower is better). The error bars show 95% bootstrap confidence intervals for average performance. We average each method over 4 seeds. Our method (blue) outperforms prior methods (orange, green). Compared to R3L [54] (orange), our approach converges ∼28.6% faster. Our ablations show that the most important aspect of our approach is having multiple resetters. This indicates that reset state convergence is crucial to good performance, in contrast to prior methods [13] that learn a explicit reset policy.

position and orientation of the object compared to the goal position evaluated from a set of initial positions that is consistent across all methods.

To evaluate our second experimental hypothesis, we evaluate skills acquired with reset-free learning for an ant locomotion task on two hierarchical navigation problem, shown in Figure 3. In these experiments, the agent must first learn a set of skills (without resets) to control an `Ant` quadrupedal robot to walk to the center of the workspace (far left), and subsequently transfer those skills to a waypoint navigation task and a maze traversal environment, which we refer to as `Ant-Waypoint` and `Ant-MediumMaze` (center, right plot respectively). For all methods, we report a return, which we normalize using the return of an agent that successfully solves the task and a random agent.

**Comparisons and baselines.** We compare our approach to prior methods, which are trained either with or without resets. In the `DClaw` environments, we compare to two prior methods that learn reset controllers: leave no trace (LNT) [13], and the R3L perturbation controller (R3L) [54]. In order to fairly compare all prior methods, we use the same underlying RL algorithm, soft actor-critic (SAC) [22], across all methods, and use the same exploration bonus used in R3L [54], random network distillation [7], which is added to the reward for all methods.

We also compare to prior skill learning methods on the `Ant` task: Diversity is All You Need (DI-AYN) [14] and Dynamics-Aware Discovery of Skills (DADS) [40]. DADS and DIAYN are unsupervised skill learning procedures that aim to learn a diverse set of skills in an environment. For this domain, prior work has differed in the choice of state representation, either using the full state [14] or a reduced $(x, y)$ coordinate state representation. For our experiments, we use the default state representation, without adding prior knowledge about the importance of $(x, y)$ coordinates. This setting is generally more challenging. We then train a hierarchical controller using the discrete skill representation using DDQN [46]. In the case of DADS [40], which uses continuous skills, we discretize the resulting skill space to allow us to use the same hierarchical learner.

**Reset-free learning results.** The results in Figure 4 show that our approach substantially improves reset-free performance. In contrast to prior approaches (orange and green), our method converges approximately 200 iterations faster on the most challenging task (`DClaw-ManipulateFreeObject`), which is a 28.6% improvement. We hypothesize that is due to the improved state coverage of our

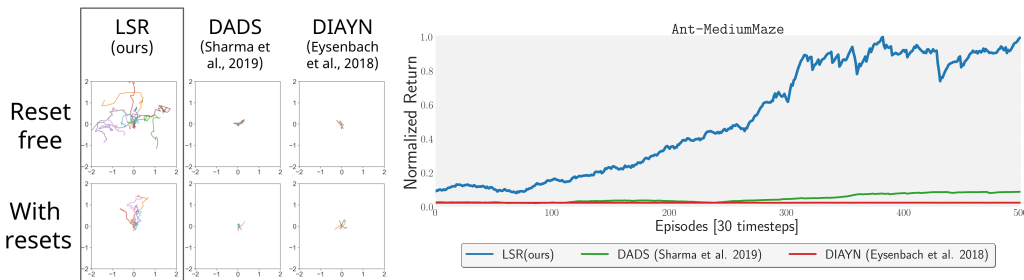

Figure 5: A visualization of the learned skills (left) and performance on a hierarchical downstream locomotion task (right). We visualize the $(x, y)$ trajectories of learned skills (left) comparing the reset-free and reset-based setting. In contrast to prior approach, LSR is able to learn skills that are able to move the ant much further from the origin in both settings. This allows us to substantially improve performance in a downstream hierarchical locomotion task (right), where a meta-controller using only the skills is able to navigate a maze much more effectively.

approach, as an explicit reset policy (green curve) [13] performs much worse. This is supported by our ablations, which show that having "Multiple-Resetters" (red), where we remove the diversity bonus, accounts for much of the method's improvements. Using a single adversary (purple-curve), where we run our approach with a single skill, performs comparably to R3L, and substantially worse than our method. Note that prior techniques that have formulated adversarial games, such as asymmetric self play (ASP) [42], generally use a single adversary.[3]

**Skill learning and hierarchical control results.** In the `Ant` domain, our approach allows us to learn reset skills that represent meaningful and useful primitives. We visualize the $(x, y)$ trajectories of the learned reset skills qualitatively in Figure 10 (left), along with skills learned by prior methods with and without resets [14, 40]. Note that the discriminators for all methods use the *full* state, but we show only the $(x, y)$ coordinates for visualization. Both prior methods (DIAYN and DADS) learn short directional walking skills with resets, similar to

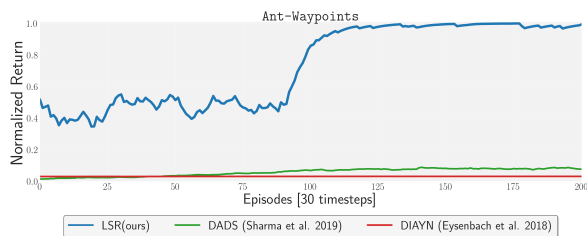

Figure 6: Performance on the `Ant-Waypoints` environment compared to DADS and DIAYN. DADS and DIAYN makes small progress on this simpler domain (orange), but our method (blue) is still able to successfully navigate between the waypoints more quickly.

those reported in prior work [14, 40], but generally fail to learn meaningful skills in the reset-free setting. However, the skills learned by our method are capable of moving the ant much further than prior methods that learn in *either* the reset-free or reset-based setting, as illustrated in the figure. Finally, we see that the skills produced by our approach improves performance in the simpler `Ant-WayPoints` navigation domain Fig. 6, and these improvements are much more pronounced in the more challenging `Ant-MediumMaze` task, where our approach leads to substantially more effective downstream learning, as shown in Fig. 10 (right), compared to prior methods. A visualization of the path taken through the maze by our hierarchical policy is shown in Fig. 3 (far right).

## 6 Discussion, Limitations and Future Work

We proposed a method that simultaneously addresses two challenges faced by real-world RL systems: learning continually without manually provided resets, and acquiring diverse skills for solving long-horizon downstream tasks. While these two problems may at first appear unrelated, we show that learning diverse skills actually enables more effective reset-free learning, while learning the skills in an adversarial manner with a task policy actually makes the skills themselves more effective, and therefore better suited for downstream long-horizon problems. Our work assumed environments that are reversible, which is very strong assumption assumption in the real world. We also assumed a

defined forward task, whereas a fully autonomous real world agent would need to have the ability to verify its own task success. A particularly exciting direction for future work would be to extend the reset game to learn a zero or few-shot success classifier, or design a system that explore conservatively and request limited human intervention when encountering a non-communicating state.

## Broader Impact Statement

The central goal of reinforcement learning is to allow agents to autonomously acquire complex behavior from interacting with the world. The central focus on this work is to address challenges in the deployment of reinforcement learning in the real world, where all learning is continual and sample efficient is important. We introduce our approach to remove the assumption of an oracle reset function and to allow the agent to simultaneously acquire temporal abstraction. In order to maximally benefit from deploying reinforcement learning agents in the real world, these challenges must be address by one method or another.

Nevertheless, as the underlying goal of our research in reinforcement learning is to enable the automatic acquisition of complex behavior, we consider the societal impact of our research to be closely related to automation. While the broader impact of having increased automation is difficult to predict, it is likely to have both negative and positive consequences. Having agents that can be deployed in unstructured environments could have the potential to automate dangerous tasks, but also could be result in more complex consequences, including changes in the employment landscape, and potential for harmful applications such as autonomous weapons.

## Acknowledgments and Disclosure of Funding

We thank the anonymous reviewers, whose comments greatly improved this work. We also thank Parsa Mahmoudieh, Michael Chang, Justin Fu and Ashvin Nair for feedback on an earlier draft of this paper. We additionally would like to give a special thanks to Justin Yu for clarifying details and helping reproduce the R3L experiments. This research was supported by the National Science Foundation under IIS-1700696, the Office of Naval Research, ARL DCIST CRA W911NF-17-2-0181, Berkeley DeepDrive, and compute support from Google.

## Footnotes

[3]While ASP shares many properties with our method, such as the use of a game formulation, it was infeasible to compare on these tasks due to differences in assumptions: ASP does not consider reset-free learning, does not aim to optimize a given task, and requires goal-reaching policy formulations for both players. However, we can view the "Single Adversary" baseline as the closest to ASP in our experiments.

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
