[Supplementary Material]

# Appendix

We first describe the details of environments used in the paper, and next list the hyperparameters used to train all the agents. All environments use the MuJoCo 2.0 simulator.

## A    Environment Details

### A.1    Ant Task

The `Ant` environment is equivalent to the standard gym `Ant-v3` environment except that the gear ratio is reduced from $(-120, 120)$ to $(-30, 30)$. The use of this lower gear ratio is consistent with prior work [14]. The observation space is the environment $qpos$ and $qvel$. The rewards are a weighted combination of three terms: (1) a negative reward of the distance from the center, (2) a positive reward for reaching a threshold from the center and (3). The equation used is for (1) and (2) is given below:

$$r(d) = e^{-d^2/2} + \text{num goals completed}$$

where $d$ is the $\ell - 2$ norm of the distance from the origin. A negative reward $(-300)$ is also given to the agent for flipping over. If the reset policy terminates the episode in this manner then the forward policy is not executed. If however, the forward policy terminates the episode, the forward policy receives the flip cost and the reset policy includes the negation of this cost in its reward as described in Sec. 4.2. Each agent is given a time horizon of $T_{\text{reset}} = T = 200$. In this ant task, the game was run for 7000 episodes.

### A.2    Ant-Waypoints Task

The aim of this task is to navigate between a set of pre-defined waypoints. The waypoints defined are $[(0, 5), (5, 5), (5, 0)]$. The reward structure is the same as the constructed ant environment described in section A.1, conditioned on the current waypoint.

The lower level controller was constructed from the reset and forward skills learned from the adversarial game. The $(x, y)$ position of the state were re-normalized so that the ant appeared to be at the origin. The higher level controller was trained using an implementation of Double DQN [46] using the 2-D coordinates of the Ant as the state.

### A.3    Ant-MediumMaze Task

We use the base environment from D4RL using a single goal[4] [19]. The reward is the $\ell - 2$ distance to the target position. The target position is the top right corner and the bottom position is the bottom left corner. Each episode is of a length 30 where each step represents a single skill.

### A.4    ManipulateFreeObject Task

The object is a 6 DoF three pronged object that is 15cm in diameter placed in a workspace that is 30 cm $\times$ 30 cm box. We follow the goal configuration of Zhu et al. [53], using $(x, y, \theta) = (0, 0, -\frac{\pi}{2})$, and use the same 15 initial configurations used for evaluation. Experiments were performed using a state representation that comprised of the object $(x, y)$ position, and $\sin$ and $\cos$ encoding of its orientation, the claw's pose, and the last action taken.

## B    Additional Experiments

## B.1 Understanding the effects of resets and state representation choice on downstream performance

Here, we provide an analysis of the effects of resets and state representation choice on downstream task performance. We vary the condition under which the policy is learned (i.e., reset or reset-free) and the state representation (full state versus $(x, y)$-prior). In general, we find that only with the availability of the $(x, y)$-prior and resets are prior methods able to make progress (Fig. 8 and Fig. 9). The different relative performance of the same actions representation on different downstream tasks also suggests that learning a single action representation may not always be ideal. This suggests future work that combines downstream adaptation, or mixing in primitive actions to allow for more granular control.

Figure 7: A visualization of skills learned when varying the state representation ($(x, y)$-prior or full state) and initial state distribution (reset-free or with resets). We find that with resets, prior method are able to produce diverse skills comparable to LSR. In the reset-free setting however, using either state representation we are able to produce superior skills when compared to prior work.

Figure 8: A evaluation of the skills learned when varying the state representation ($(x, y)$-prior or full state) and initial state distribution (reset-free or with resets) on the `Ant-MediumMaze` task. We normalize the return of plot in terms of the best performing algorithm. We find that with the $(x, y)$-prior is necessary to allow prior methods to perform comparably to LSR. In the reset-free learning setting using full-state however, only LSR approach is able to successfully traverse the maze.

Figure 9: A evaluation of the skills learned when varying the state representation ($(x, y)$-prior or full state) and initial state distribution (reset-free or with resets) on the `Ant-Waypoints` task. We normalize the return of plot in terms of the best performing algorithm. Similar to the `Ant-MediumMaze` task, we find that prior methods can only make progress on the task when resets or additional information in the form of the $(x, y)$-prior is available.

## B.2 Effect of varying the number of skills on downstream performance

Figure 10: A plot showing the effect of varying the number of skills on the `Ant-Waypoints` tasks. We find that increasing the number of skills improves performance (16, yellow curve), the value used in our experiments (10, purple curve) is far from the best, and many other values attain similar performance. The number of skills used in our experiments were chosen to be roughly around the order of previous work [14].

# C   Algorithm Details

The adversarial game comprises of a reset policy and a forward policy as described in the paper. The forward policy receives rewards as given by the environment at every step. The reset policy receives rewards from DIAYN at every step and a game reward at the last step. This game reward is the sum of the rewards received by the forward policy when played from the final state of the reset policy trajectory. Each agent is given a fixed number of steps at each turn. Environment termination was handled for individually for both agents as follows. If the reset policy terminates, the forward policy does not take any steps in the environment and is not given any reward. If forward policy terminates, the reset policy is given rewards based on the forward policy as normal.

In the following sections, curly brackets indicate that a parameter was searched over.

## C.1   DClaw Environment Parameters

The experiments conducted on the `DClaw-ManipulateFreeObject` domain was conducted using the following parameters. The grid search parameters for DIAYN were identical to those used in our method except where $\lambda = 0$.

| Hyperparameters | Value |
|---|---|
| Actor-critic architecture | FC(256, 256) |
| RND architecture | FC(256, 256, 512) |
| Classifier architecture | FC(256, 256) |
| Optimizer | Adam |
| Learning rate | {3e-3, 3e-4} |
| Classifier steps per iteration | 5 |
| $\gamma$ | 0.99 |
| $\lambda$ | {0.1, 0.5} |
| $r_{\text{skill}}$ scale | 0.1 |
| target update | 0.005 |
| Batch size | 256 |
| Classifier batch size | 128 |

We follow [54] and provide 200 goal images for all experiments which are used to learn a VICE [18] based reward function. We similarly omit the $-\log \pi(\mathbf{s}|\mathbf{a})$. We also follow Burda et al. [7] and normalize the prediction errors. In our implementation of LSR, we found used a dimensionality of $\mathbf{z}$ of 2, implemented as a separate network as opposed to one network. The reset controller [13] and R3L baseline [54] follow the hyperparameters above. The reset controller's results were averaged over the 3 positions suggested by Zhu et al. [54].

## C.2   Ant Environment Parameters

The experiments conducted on the `Ant` domain were conducted using the following parameters. The grid search parameters for DIAYN were identical to those used in our method except where $\lambda = 0$.

| Hyperparameters | Value |
|---|---|
| Actor-critic architecture | FC(256, 256) |
| Classifier architecture | FC(256, 256) |
| Optimizer | Adam |
| Learning rate | {1e-3, 2e-4} |
| Training steps per agent | {200, 200} |
| $\gamma$ | 0.99 |
| Episode Length | 1000 |
| $\lambda$ | 0.5 |
| Number of skills | 10 |
| $r_{skill}$ scale | {0.01, 10}, chose 2 |
| target update | 0.005 |
| Batch size | 256 |
| Classifier batch size | 256 |

## C.3   Hierarchical Controller Parameters

The hierarchical controller is a Double-DQN [46] network that takes as input the $\mathbf{s}_t$ and outputs a Q-function over the skills $Q(\mathbf{s}_t, \mathbf{z})$.

| Hyperparameters | Value |
|---|---|
| Architecture | FC(128) |
| Exploration fraction | 0.3 |
| Final epsilon j | 0.0 |
| Learning rate $\alpha$ | 0.0001 |
| Reward Scale | {0.1, 0.3} |
| Batch size | 256 |
| Maximum horizon (T) | 30 |
| Total epochs | 500 |
| Replay Buffer size | {1000, 10,000} |
| Critic update frequency | {5, 10} |
| Critic updates per epoch | {100, 200} |

An episode step corresponds to a 150 length rollout by the lower level controller in the environment.

## C.4 DADS

To allow for fair comparison, we run DADS without an $(x, y)$ prior. We additionally discretize the continuous skill space into a space of 10 discrete skills to allow for us to use the same meta-controller architecture. We used the open source implementation of DADS [40] [5] with the following parameters:

| Skill parameters | |
| --- | --- |
| Number of skills | 10 |
| Skill type | Discrete |
| **Dynamics parameters** | |
| train steps | 8 |
| learning rate | 3e-4 |
| batch size | 256 |
| Path length | 200 |
| **Agent parameters** | |
| Architecture | FC(512) |
| Learning rate | 3e-4 |
| Entropy | 0.1 |
| Train steps | 64 |
| Batch size | 256 |
| $\gamma$ | 0.99 |

## Footnotes

[4] https://github.com/rail-berkeley/d4rl

[5] https://github.com/google-research/dads