[Reviews · NeurIPS 2020]

Review 1

Summary and Contributions: One of the barriers to using RL in realistic settings is the need to reset the environment. In addition, discovering generally-useful skills is a longstanding goal of unsupervised RL. This work contributes an elegant solution to both problems by treating learning to reset and skill discovery as a single objective in a two-sided general-sum game. The work provides a thorough motivation of the approach, including a brief theoretical characterization, and provides experimental validation of its use in reset-free learning and as a skill discovery method for use in downstream hierarchical RL.

Strengths: This work addresses seemingly disparate challenges (reset-free learning and skill discovery) in an interesting and novel way, creating a bridge between them via a form of asymmetric self-play. The authors provide a sound game-theoretic characterization, which helps to inform some of their methodology. The authors empirically demonstrate that their method outperforms other reset-free learning approaches on a challenging benchmark task. They also show that skill discovery in the context of their resetting game yields skills that offer a better action space for a high-level controller when applied to a downstream hierarchical RL task. This work should be generally relevant to RL researchers within the NeurIPS community. While the underlying concepts that this work builds on are not novel, their combination introduced by this work is novel.

Weaknesses: While the theory is laid out clearly, there are a handful of practical considerations that don't receive sufficient attention. In particular, how to choose the task for the forward policy in the resetting game. Perhaps this is obvious when the goal is simply to do RL in a reset-free environment (as in, I believe, Figure 4), but it's not obvious when the goal is doing skill discovery. At the very least, it would be useful to acknowledge environments where the proposed resetting game would not be feasible. Very little effort is given to characterizing the learned skills. This is likely an issue of space restriction, but no such analysis is provided in the appendix either. There is also some concern that the downstream HRL experiments are somewhat unfair to the baseline methods (DADS and DIAYN), which are known to perform worse when the discriminator receives the full state instead of (x, y). The authors' implementation choice potentially turns those baselines into "straw men", weakening the authors' claims around Figures 5 & 6.

Correctness: The appendix describes some heuristics that the authors used for adapting the policies learned during the resetting game to the downstream HRL tasks. At face value, these heuristics seem relatively minor, but they raise the concern that the authors produce an unfair comparison--especially if these heuristics significantly change the results. At the very least, the importance of these choices should be presented (even if only in the appendix) and the authors should be clear about any consistent difficulties faced when adapting the policies to downstream HRL.

Clarity: Yes, very well written.

Relation to Prior Work: Yes, for the most part. The paper could be improved by adding more discussion around reset-free learning. However, because this paper intersects 3 topics in RL (reset-free learning, unsupervised RL, and asymmetric self-play), it is forgivable that the authors choose to be brief.

Reproducibility: Yes

Additional Feedback: I will be happy to raise my score if the authors can address my concerns about: - The full-state vs (x, y) discriminator input in the DADS & DIAYN baselines. - The heuristics described in the appendix. Minor details: - There appears to be a typo in the legend of Figure 4. The purple line is labeled "No Diversity Bonus" but the main text provides a different description. - In line 233 "experimental questions" would be more appropriate phrasing than "experimental hypotheses." (A hypothesis is a statement, but what follows in the text are questions). POST-REBUTTAL COMMENTS: In rebuttals, the authors have provided some results for the with- and without-(x, y) variants and show that the comparison results do not appear to change dramatically. I think their comparison to DADS & DIAYN is much stronger for including both variants. In the discussion phase, the authors' communications with the AC were made available to the reviewers; in this communication, the authors make a compelling argument that reviews should not punish authors for describing implementation details (even if they may be perceived as "tricks"). The authors make the good point that reproducibility is harmed when authors are incentivized to simply obscure those details under the hopes that reviewers won't notice. If, in the final paper, the authors end up pointing out these details and also providing some discussion on their importance (which it sounds like they will do), that strikes me as responsible. I agree with the overall sentiment that this paper is high quality and I think it's a good candidate for publication. I have increased my score accordingly.


Review 2

Summary and Contributions: POST REBUTTAL EDIT: Thank you to the authors for the responses. My review and score stands reinforced. This paper considers the seemingly disparate ideas of learning in reset-free settings and learning useful abstract skills together, showing that they can benefit from each other. Reset-free settings more closely resemble real-life learning where resetting after each episode may be too time-consuming or impossible. The agent should learn to automatically reset itself to challenging start states so it can attempt the task again. Learning abstract skills is important for effectively exploring hard environments and for solving downstream hierarchical tasks. The insight linking the two ideas is to set up a two-player game such that a regular task policy learns to solve the task, while a skill policy is simultaneously learning to reset the agent to starting states. The game executes in turn, with the reset policy receiving the negative reward cumulated by the task policy to encourage it to reset to states that are challenging to solve the task from. This sets up an automated curriculum for learning as the task and reset policies improve simultaneously. Crucially, the reset policy is also encouraged to lead to diverse starting states by conditioning on a latent skill vector and adding a variational bonus to its reward. The effectiveness of the approach is proven on two reset free domains (claw manipulation and Mujoco ant) where it outperforms recently published prior work and on a hard hierarchical task where it also substantially outperforms prior work. Overall, this is an excellent paper that links two seemingly different ideas together with a clever insight!

Strengths: + The contributions in this paper are novel as far as I am aware. Prior work treats resetting and learning of abstract skills separately whereas in this work the link between the two is exploited successfully. + The work addresses an elephant in the room for RL - how do you reset environments outside of simulation where it may be time-consuming or costly to do so frequently. The proposed approach is to allow the agent to learn diverse skills to push itself to challenging state states and use these skills for downstream task execution as well. + There is some evidence that the skills learned by forcing the agent to reset to challenging start states for a task may be more diverse than the previous SOTA that learn skills unsupervised-ly. This is an exciting direction of work for the NeuIPS community! + The approach is adequately tested on two challenging environments and against recent baselines and ablations of the approach.

Weaknesses: Overall this is an excellent submission. Congratulations! The following is an attempt to find some weaknesses as this section requires me to: - Not all RL environments can be “reset” by reaching a state from which the task execution is challenging. For example, pouring water into a container can be very challenging if the agent spills the water on the floor first. What are your thoughts on the limitations of the approach and in general, safe learning while also trying to find challenging states to reset to? I suggest some discussion of this in the paper. - The ablation conditions could be explained more. I am still a bit unsure what the “Multiple-Resetters” ablation is exactly. - Did you attempt any other hierarchical tasks, within the ant environment or on the other Mujoco ones?

Correctness: The claims and methodology are correct to the best of my knowledge.

Clarity: The paper is clear and well written! Small typos are present which can be addressed in the camera-ready version. Line 91: agents -> agent Line 183: “due to the fact” repeated Line 236: ?. -> ? Line 248: experiment -> experiments Line 314: efficient -> efficiency

Relation to Prior Work: Related work was adequately discussed and distinction from prior work was established due to the novelty of the approach.

Reproducibility: Yes

Additional Feedback: I could not find code with the submission. The paper contains enough details to be able to reproduce the experiments but I would encourage the authors to open source the code. In conclusion, a novel, interesting, technically sound approach that links together two seemingly disparate ideas of reset free learning and learning abstract skills.


Review 3

Summary and Contributions: A often ignored problem when developing reinforcement learning algorithms is that any agent deployed in the real world that encounters a new problem will likely not have access to human supervision. To learn to any degree of competency, the agent will need to attempt the problem many times and thus will require some mechanism by which to reset the problem to an initial state. This manuscript under consideration proposes the Learning Skillful Resets (LSR), a methodology for allowing an RL agent to learn to reset a problem to, challenging, initial states. LSR formulates this resetting procedure as a Stackelberg game in which two policies (a reset policy and a task policy) compete. Their empirical results show that can improve sample learning effiency in this reset-free domain and that the skills learned by the reset policy can be used downstream to solve other tasks using hierarchical reinforcement learning. #### POST-REBUTTAL RESPONSE #### This paper has obtained broad consensus, it makes interesting, well-evaluated, methodological contributions and is well-written. The author's rebuttal (and response to the AC) has presented several new interesting experimental results (e.g. including the (x,y) prior to more comprehensively compare against other HRL methods) and addressed the other minor comments I had. While I cannot imagine it will have any impact on this paper's acceptance, I have increased my final score to recognize the author's efforts in this rebuttal process.

Strengths: The problem tackled by this work, reset-free learning, is important and under-explored. The authors' LSR methodology is well-formulated and leaves enough technical details to encourage future (possibly theoretical) follow-up work. The experiments validating the approach, as well as the results of using the reset policy within hierarchical controllers, are promising if somewhat limited.

Weaknesses: # Supervision for goal reaching As one of the primary motivations appears to be real-world realism it seems reasonable to ask: is it reasonable to assume that an RL agent deployed in the real-world will have the ability to consistently verify that it has successfully completed it's task? E.g. in your ant locomotion task, we must be able to verify that the agent has returned to the workspace's center. Perhaps it's possible to learn a zero-shot classifier for task completition but this seems like a significant obstacle to learning in the real world. While I don't necessarily expect a solution to this problem, some discussion seems reasonable. # Empirical validation While the experiments seem well focused, the claim of 28.6% faster convergence is a bit suspect. How did did you control for hyperparameter tuning? I would have also liked to see this faster convergence claim validated on more tasks: can we have a similar plot as in Fig. 4 for the ant task?

Correctness: Yes, the claims, methods, and empirical methodology appear correct.

Clarity: Yes, the paper was very well written and an enjoyable read.

Relation to Prior Work: The authors appear to have done a sound job comparing to prior work.

Reproducibility: Yes

Additional Feedback: Line-by-line comments: Line 129 - Seem to be missing a comma after the $\ldots$. - Perhaps also add a subscript of $\gamma$ to $R$ to emphasize the dependence? Line 177 - One theoretical point that seems hidden or ignored in this work is what this expectation for $J^{forward}$ (and $J^{reset}$) really means. Because of the iterative and continuous "reset, forward, reset, forward, ..." nature of the task, this expectation is being (implicitly) taken after some arbitrary number of iterations between resets and forward episodes. This is perhaps fine if the initial states converge to some non-degenerate stationary distribution but this ignores the, very real, possibility of there being inescapable terminal states. E.g. if the reset policy always eventually throws the robot into a hole then the stationary distribution will always have the robot in this hole and thus nothing can be learned. Perhaps add an assumption that the underlying MDP is irreducible? Lines 182-184 - Looks like there was a copy-pasting error here. Lines 185-186 - I would like a little more detail regarding what "solvable" means here. Does this just mean a gradient based method will converge to something (but not something that is necessarily optimal)? Equations (4) and (5) - As with my comment for line 177, I'm not sure this formulation is exact. Line 236 - Extra period after a question mark. Figure 4 - Are you simply bootstrapping 4 values at every timepoint here? At this point, why not just show the minimum and maximum performance? - There is an error in the display of 28.6%, I believe you wanted to use $\sim$ rather than $\tilde$. - Last line, "converge" should be "convergence" Lines 273-274 - What does "improved state converage" really mean here? Can you visualize a post-reset heatmap of object locations. If I take the distribution of object locations taken in this way and train a new reset-allowed variant with this distribution, will I obtain equivalent/better performance? I'm curious here if the reset policy is implicitly performing curriculum learning during training or if it's just the final reset states that are important. Figure 5 - "skills that able to" -> "skills that are able to" Lines 315-316 - "If we are to gain any..." This seems unduly pessimistic.


Review 4

Summary and Contributions: The paper proposes a method for non-episodic reset-free learning, evaluated in simulated environments, which uses unsupervised skill discovery to learn a diverse set of "reset skills" that are used by a reset policy in order to replace manual/oracle environment resets, and a main task policy that learns to optimize task performance after starting from the states at which the reset policy terminates. Experimental results show the importance of all components of their objective - specifically their diversity term in unsupervised reset skill learning, which leads to the most improvement over baselines that use a single reset policy. ================= Post-rebuttal update: Having read through the concerns of other reviewers and the author's response, I am happy that most concerns were addressed (especially about the fairness of comparisons with DADS/DIAYN), except the concern about the significance of the "28.6% improvement" in convergence rate. Nevertheless, I recommend that this paper be accepted and trust that the authors will add the missing concerns for the final revision. I have increased my score to reflect my recommendation.

Strengths: - The paper presents a novel method which addresses two challenges simultaneously: learning in a reset-free setting and learning diverse skills (i.e. reset skills) in an unsupervised manner. - The diversity term in the proposed objective (which is the main novelty of the paper over a single reset policy baseline) is shown to be the reason for improved performance via the ablations in Fig 4. - The skills learnt by the reset policy are shown to be useful primitives in a separate hierarchical RL task (in the Ant env). - When comparing just the unsupervised skill discovery part of the proposed method, a clear improvement in the skills is shown (in Fig 5) over prior work (i.e. DADS, DIAYN) in terms of how far the skills reach from the origin for the Ant locomotion task.

Weaknesses: - The curves shown in Figure 4 does show an improved convergence rate (28.6% as mentioned in the caption) w.r.t the single reset policy baseline of prior work, but just the number does not convey whether this is a significant improvement for practical purposes. Given that a reset-free task does not require human-assisted reset for say, a real robotic experiment, is this really a significant boost given that the more complicated (and potentially more difficult to learn) objective i.e. the added diversity term? I would be more convinced with experiments that show the stability of the proposed objective, given that it is adversarial in nature while also enforcing the learning of a set of unsupervised skills. - The supplementary material mentions that 2 skills were learnt with DClaw and 10 skills were learnt with the Ant env. The importance of this hyperparameter is not mentioned in the paper, nor is the process of selecting it. Also, are all 10 skills learnt in the Ant environment distinct? Is there a curriculum in number of skills learnt over training? - L205 mentions that hyperparameter \lambda being set to 0 reduces to a prior skill learning method (e.g. DIAYN). It seems like this setting can also be a valid candidate for a reset-free learning method, and should have been compared with in Fig 4.

Correctness: The claims, methodology and empirical results are correct.

Clarity: Yes, the paper is well written.

Relation to Prior Work: Yes, prior work is adequately discussed and brought forward in empirical comparisons.

Reproducibility: Yes

Additional Feedback: Major errors: - Legend in Figure 4 seems to be incorrect, as the L275 mentions that the "Multiple-Resetters" (red line) corresponds to removal of diversity penalty whereas the "Single Adversary" (purple line) corresponds to a single skill. The figure legend's purple line label should corrected to "Single Adversary". Minor errors: - L183: Typo - "Due to the fact..." is repeated. - L205: "\lambda controls the relative importance of the two terms of J^{forward}" -> this should be corrected to J^{reset} since it has two terms. - Fig4: "This indicates that reset state converge is crucial" -> convergence

[Author Response · NeurIPS 2020]

*We thank the reviewers for the thoughtful feedback and suggestions. We address specific questions and include*
*suggested evaluations. Any clarifications that were raised and not specifically discussed here due to space will be*
*addressed in the final version, and we will include all requested clarifications and discussion of prior work.*

**R1 – "some concern that the downstream HRL experiments are somewhat unfair to the baseline methods**
**(DADS and DIAYN)"** We emphasize that the experiment is set up in a fair way, in the sense that all methods get
the same state representation. That said, we agree that DADS and DIAYN are capable of performing better with
additional supervision using the (x,y) prior (as is our method!). To address this concern, we present results below to help
characterize the different skill learning capabilities across different settings. We will include quantitative comparisons
for all of these in the final paper.

All methods learn meaningful skills, but DADS skills with the $(x, y)$ prior travel much further. LSR learns skills that
travel far both with and without the xy prior, and LSR learns substantially better skills in both settings in the reset-free
setting. We will also investigate other design choices suggested by R1 in the final version of the paper.

**R1,R3 – "how to choose the task for the forward policy" / "is it reasonable to assume that an RL agent deployed**
**in the real-world will have the ability to consistently verify that it has successfully completed it's task"** We agree
that this is an important real world challenge that our approach does not attempt to address. We will highlight this in the
final version as an important challenge for building a full real-world learning system. Multiple prior works do already
study these questions, including methods for learning rewards and verifying the success of a task, learning tasks from
demonstrations, etc. With regards to the interesting suggestion of learning a zero or few-shot classifier, it would be
interesting future work to investigate how prior [Xie et al., CoRL 2018] could be adopted or made compatible to the
assumptions in our work.

**R2,R3: "Not all RL environments can be "reset" by reaching a state" / "Perhaps add an assumption that the**
**underlying MDP is irreducible"** These comments raise good points which we will address by adding a "Limitations
and Future Work" section to the final version of the paper. In short, the reset-free learning setting that we consider
implicitly assumes that the underlying MDP is resettable from different states. This is equivalent to limiting our approach
to MDPs which are irreducible. Prior work on reset-free learning has also made this assumption [Chatzilygeroudis et
al. RAS 2016, Eysenbach et al. ICLR 2019, Zhu et al. ICLR 2020]. Interesting future directions to address this issue
would be design systems that explore conservatively (to avoid scenarios such as ones described by R2) or make use of
limited human supervision in non-reversible MDPs when the agent encounters a non-communicating state.

**R4: 'the stability of the proposed objective'** We found that our
approach is reasonably stable across different seeds in our method,
which we plot individually in the plot on the right. We will add a
detailed hyperparameter stability analysis in the final version, as we
did not have time to complete one during the rebuttal phase.

**R3: "did you control for hyperparameter tuning"** We followed the
base hyperparameter exactly as provided in prior work [Zhu et al. ICLR
2020], which the authors tuned specifically for their approach. We did
not re-tune these base parameters beyond the initial learning rate and
tuned the scale term $\lambda$ introduced in our approach using a gridsearch.

**R4 – "The importance of this hyperparameter is not mentioned"**
In the paper, we chose this hyperparameter to be roughly on the order
of prior work that considered similar domains. We agree however, that
this is an important hyperparameter, as it can have a complex effect on
downstream RL performance. We provide on the right a plot showing
the effect of varying the number of skills on the `Ant-Waypoints` tasks.
We find that increasing the number of skills improves performance (16,
yellow curve), the value used in our experiments (10, purple curve) is
far from the best, and many other values attain similar performance.



[Meta-Review · NeurIPS 2020]

The rebuttal and discussion phase managed to convince all reviewers that this paper definitely should be published (see updated reviews). The paper proposes a very interesting method and also backs the idea up with thorough evaluations, coming up with an innovative link between two problems that to date had been treated separately.